# Antiarrhythmic Drug Dosing in Children—Review of the Literature

**DOI:** 10.3390/children10050847

**Published:** 2023-05-08

**Authors:** Nathalie Oeffl, Lukas Schober, Patrick Faudon, Sabrina Schweintzger, Martin Manninger, Martin Köstenberger, Hannes Sallmon, Daniel Scherr, Stefan Kurath-Koller

**Affiliations:** 1Division of Pediatric Cardiology, Department of Pediatrics, Medical University of Graz, 8036 Graz, Austria; 2Division of Cardiology, Department of Medicine, Medical University of Graz, 8036 Graz, Austria

**Keywords:** children, antiarrhythmic drugs, arrhythmia, electrophysiology, drug dosing

## Abstract

Antiarrhythmic drugs represent a mainstay of pediatric arrhythmia treatment. However, official guidelines and consensus documents on this topic remain scarce. There are rather uniform recommendations for some medications (including adenosine, amiodarone, and esmolol), while there are only very broad dosage recommendations for others (such as sotalol or digoxin). To prevent potential uncertainties and even mistakes with regard to dosing, we summarized the published dosage recommendations for antiarrhythmic drugs in children. Because of the wide variations in availability, regulatory approval, and experience, we encourage centers to develop their own specific protocols for pediatric antiarrhythmic drug therapy.

## 1. Introduction

Supraventricular tachycardias (SVTs) represent the majority of arrhythmias in pediatric patients, including the foremost atrioventricular reentry tachycardia (AVRT), atrioventricular nodal reentry tachycardia (AVNRT) and atrial tachycardia (AT). Atrial fibrillation (AF) and atrial flutter (AFL), predominating in the adult population, are less frequently encountered in pediatric patients. AF and AFL occur mainly following heart surgery or, occasionally, may be caused by alcohol consumption [1,2]. Ventricular tachycardia (VT) and ventricular fibrillation (VF) are rare among pediatric patients [2,3].

Thus, target arrhythmias for antiarrhythmic drug (AAD) therapy in pediatric patients differ from those in adults. Among the substances used, betablockers and class I antiarrhythmics dominate. Other substances, e.g., ivabradine, are reserved for exceptional cases only. Recommendations with regard to indications and dosage in pediatric patients are limited and based largely on expert consensus [2,3,4]. In clinical practice, however, dosage recommendations from medical books and online platforms or applications are broadly accessed and frequently employed. For some drugs, substantial gaps with regard to dosing recommendations exist, and comprehensive information on pediatric patients remains scarce. In this article, we aim to provide a review on dosage recommendations for AADs in the pediatric cohort.

## 2. Materials and Methods

We conducted a structured review with regard to dosage recommendations for commonly used antiarrhythmic agents in the pediatric cohort. A total of 18 antiarrhythmic substances commonly used in pediatric patients in central Europe were considered. Included among these is ajmaline, which is frequently used in Germany and Austria but is not ubiquitously available.

We searched PubMed, MEDLINE, and CINAHL using the following terms in different combinations: “pediatric”, “children”, “antiarrhythmic drugs”, “antiarrhythmic therapy”, “arrhythmia treatment”, and “antiarrhythmic substances”. Papers with regard to AAD dosing in out-of-hospital resuscitation, recommendations on AAD dosing in resuscitation scenarios only, or dosing recommendation for other causes than arrhythmia therapy were excluded. Every study was assessed independently by two reviewers.

Furthermore, we considered the following online resources:

Prescription drugs—information, interactions and side effects, Hazleton, PA, USA (drugs.com) accessed on 10 April 2023.

Antiarrhythmic drug guide, Department of Pediatrics, School of Medicine, Washington University in St. Louis, MO, USA (pediatrics.wustl.edu/cardiology/antiarrhythmic-drug-guide) accessed on 10 April 2023.

We included guideline papers and expert consensus statements on antiarrhythmic therapy in children by the European Heart Rhythm Association (EHRA) and the Association for European Pediatric and Congenital Cardiology (AEPC) [2,3].

Data on antiarrhythmic drug dosing in children were gathered, focusing on commonly used antiarrhythmic drugs. Dosage recommendations were outlined based on body weight and/or body surface area and given as the minimum to maximum dosage.

## 3. Review on Antiarrhythmic Drug Dosing in Children

Indications for the use of antiarrhythmic drugs are provided by guidelines and consensus statement and position papers [2,3,4]. Dosage varies by indication but also between different resources of information on specific drugs and indications. Table 1, Table 2, Table 3, Table 4 and Table 5 summarize the range of dosage recommendations for each antiarrhythmic substance. Table 6 provides data on antiarrhythmic drugs, including indications, preparations, and mechanism of action.

Class 0 antiarrhythmic drugs

Ivabradine, a class 0 AAD with hyperpolarization-activated cyclic nucleotide-gated (HCN) channel blocking properties, is available for oral administration only. Dosage is reported as 0.05 mg/kg twice daily in children <40 kg body weight. It may then be uptitrated to 0.2 mg/kg per dose in infants 6 months to 1 year of age and to 0.3 mg/kg per dose (NOT per kg) in children >1 year of age. A maximum single dose is reported up to 7.5 mg (NOT per kg). In children >40 kg body weight, the initial dose is reported as 2.5 mg per dose twice daily, not to exceed 7.5 mg per dose (NOT per kg). Ivabradine is a relatively new drug, approved by the FDA in 2015 (for heart failure therapy) [100]. It has not yet been incorporated into pediatric arrhythmia guidelines, and data on its use in pediatric patients are mainly based on case reports. Ivabradine’s half-life is reported up to 6 h, with peak plasma levels within 1 h following intake when fasting and 2 h when taken with food. Ivabradine may cause negative inotropic and vasoconstrictive effects, arrhythmias, impaired intracardiac impulse propagation, or arterial hypotension. Rarely, dysphagia has been reported [5,6,7,8,9]. Case reports show beneficial effects in tachyarrhythmias, showing no increase in adverse events compared to use in adults. Future recommendations will most likely include ivabradine for tachyarrhythmia treatment in pediatric patients.

Class I antiarrhythmic drugs

Ajmaline, a class Ia AAD with sodium blocking properties, is an alkaloid from the roots of *Rauvolfia serpentia* (Indian snake root). The reported initial dosage ranges from 0.5 to 1 mg/kg iv. The maximum dosage is reported up to 50 mg per dose or 8 mg/kg per day or 1.200 mg per day (NOT per kg) for continuous infusion. If using continuous iv dosing, the initial dose is reported to range from 0.5 to 1 mg/kg per hour, not to exceed more than 10 mg/min (NOT per kg). However, ajmaline is not available in most countries, even though it is recommended for diagnostic testing in cases of suspicion for Brugada syndrome. In that case, the dosage is reported as 1 mg/kg, administered either over 10 min continuously iv or as bolus of 10 mg per dose iv every 2 min. Common indications for ajmaline include diagnostic testing for Brugada syndrome. Its use in supraventricular tachycardia is not an established practice in pediatric patients. Ajmaline’s half-life is reported to range from 6 to 96 min. Ajmaline may exhibit negative inotropic and vasoconstrictive effects, among others [2,10,11,12].

For flecainide, a class Ic AAD, the intravenous bolus dose is reported to range from 0.2 to 2.0 mg/kg per dose, administered every 12 h. Furthermore, a single bolus dose is reported to be followed by continuous infusion ranging from 0.2 to 0.5 mg/kg/h. Flecainide continuous infusion is also reported without prior bolus administration. The maximum dosage is reported up to 200 mg/day (NOT per kg). For oral administration, the dosage is reported based on body weight (BW) and based on body surface area (BSA). The BW dosage is reported to range from 1 to 7 mg/kg per dose administered every 8 to 12 h. Based on BSA, the initial dosage is reported as 50 mg/m^2^/day in children below 6 months of age. For children older than 6 months, the dosage is reported as 100 mg/m^2^/day. The maximum dosage is reported up to 200 mg/day (NOT per kg) or 8 mg/kg/day. Common indications for flecainide include supraventricular tachyarrhythmias, including Wolff–Parkinson–White syndrome. It may also be used to control hemodynamically stable wide complex tachycardias. It is also used to treat prenatal tachyarrhythmias. Flecainide’s half-life is reported to be up to 29 h in newborns, up to 6 h in infants below one year of age, and up to 8 h in children older than one year of age. Flecainide may cause dizziness, visual disturbance, dyspnea, chest pain, or proarrhythmia. The latter is reported especially for patients with Brugada syndrome and is reported as a potential fatality [2,22,23,24,25,26,27,28,29,30,31,32,33,34].

For lidocaine, a class Ib AAD, the dosage is reported as 1 mg/kg per intravenous dose, followed by continuous infusion ranging from 20 to 50 mcg/kg/min. The maximum loading dose is reported to be up to 1 mg/kg. The maximum dosage for continuous infusion is reported to be up to 50 mcg/kg/min. Common indications for lidocaine include life-threatening ventricular tachycardia or ventricular fibrillation according to current ERC guidelines, hemodynamically stable wide complex tachycardia, or drug refractory supraventricular tachyarrhythmias. Lidocaine’s half-life is reported to be up to 2 h, with the onset of action within 90 s following iv administration. There is no preparation of lidocaine available for oral administration. The effect duration is reported to be up to 20 min. Lidocaine may cause headache, shivering, bradycardia, arrhythmias, or circulatory shock, among others [2,37,38,39,40].

For mexiletine, a class Ib AAD, the dosage is reported to range from 1.4 to 5 mg/kg per dose, administered orally every 8 to 12 h. The maximum dosage is reported to be up to 8 mg/kg per dose (or 450 to 1.200 mg/day in adults). There is no preparation of mexiletine available for intravenous administration. Mexiletine is indicated in long QT syndrome type 3, where it has been shown to significantly reduce the risk for sudden cardiac death/ventricular arrhythmias. Mexiletine’s half-life is reported to be up to 12 h. The onset of action is reported as early as 30 min, ranging up to 120 min following administration. Mexiletine may cause headache, heartburn, chest pain, dizziness, visual disturbances, or nausea and vomiting, among others [13,14,15,16].

For propafenone, a class Ic AAD, the dosage is reported to range from 0.2 to 2.0 mg/kg intravenously over two hours, followed by continuous infusion ranging from 4 to 20 mcg/kg/min. The maximum dosage is reported to be up to 2.0 mg/kg iv over two hours (loading dose) and 20 mcg/kg/min for continuous infusion. For oral administration, the dosage is reported based on body weight (BW) and based on body surface area (BSA). The BW dosage is reported to range from 6 to 20 mg/kg/day, administered every 8 to 12 h. Based on BSA, the dosage is reported to range from 150 to 600 mg/m^2^/day. The maximum oral dosage is reported to be up to 20 mg/kg/day or 900 mg/day (NOT per kg, in adults). Common indications for propafenone include supraventricular tachyarrhythmias or idiopathic monomorphic ventricular tachycardia. Propafenone’s half-life is reported to be up to 32 h in poor metabolizers and up to 10 h in rapid metabolizers. Propafenone may cause unusual taste, nausea, dizziness, or arrhythmias, among others [2,24,35,36].

Quinidine, a class Ia AAD, is available in some countries as quinidine sulfate or quinidine gluconate. For quinidine sulfate, the oral dosage is reported to range from 15 to 30 mg/kg/day, administered every 6 h. The maximum dose is reported to be 60 mg/kg/day. The dosage based on BSA is not well described. BSA-based dosing of 900 mg/m^2^ is reported. Quinidine sulfate is not available for iv administration. Quinidine gluconate dosing is reported as 20–60 mg/kg orally, administered every 8 h. For quinidine gluconate intravenous dosing is reported to be 30 mg/kg/day or 900 mg/m^2^/day, administered every 5 to 6 h. Indications for quinidine’s use in pediatric patients mainly focus on Brugada syndrome-associated electrical storm or Short QT syndrome. Quinidine’s half-life is reported to be up to 8 h. The onset of action is reported within 3 h following oral administration. Quinidine may cause lightheadedness, heartburn, diarrhea, nausea, or vomiting. Torsade-de-pointes tachycardias have been reported when it is administered concomitantly with clarithromycin. When patients are on warfarin, closely monitor coagulation, as quinidine may potentiate warfarin effects. Grapefruit juice may delay gastrointestinal absorption [17,18,19,20,21].

Class II antiarrhythmic drugs

For adenosine, a revised Vaughan–Williams classification [101] class IIe AAD, the dosage is reported uniformly throughout literature resources as 0.1 mg/kg, with repetitive doses of 0.2 and 0.3 mg/kg. In children younger than one year, the initial dose is reported to be uptirated to 0.15 mg/kg. Given the nature of the drug, there is no preparation of adenosine available for continuous iv or oral administration. Common indications for adenosine include the acute treatment of supraventricular tachycardia. It may also be used for diagnostic purposes in hemodynamically stable wide complex tachycardias. Adenosine’s half-life is reported to be less than 10 s due to the rapid uptake by vascular endothelial cells and erythrocytes with rapid metabolization. Adenosine may cause chest discomfort, facial flushing, neck discomfort, gastrointestinal distress, or cardiac arrhythmia, among others [2,3,4,75].

For atenolol, a class IIa AAD, dosage is reported range from 0.05 to 0.2 mg/kg per dose. The maximum dose is reported as 2.5 mg per dose. If using continuous iv dosing, the initial dose is reported to range from 0.5 to 1 mg/kg per hour, not to exceed 10 mg/min. If used orally, the dosage range is reported from 0.3 to 1.0 mg/kg per day, with a maximum dose of 2 mg/kg per day or 100 mg per day. Common indications for atenolol include recurrent paroxysmal supraventricular tachycardias and long QT syndrome 1 & 2. Atenolol’s half-life is reported to be up to 35 h, with 16 h reported in newborns and up to 7 h reported in children aged 5 years or older. The onset of action is reported within 1 h following oral administration. Atenolol may cause arterial hypotension, fatigue, tachycardias, or bradycardia, among others [27,41].

For atropine, a class IIc AAD, the dosage is reported to ranging from 0.01 to 0.04 mg/kg per dose. For atropine, there is a recommended minimum dose of 0.05 to 0.1 mg per dose. The maximum dose is reported to be 0.5 to 1 mg per dose. There is no continuous iv dosing or oral dosage preparation available for atropine. Common indications for atropine include symptomatic bradyarrhythmias, especially associated with betablocker or calcium channel blocker toxicity. Atropine’s half-life is reported to be up to 10 h, with the longest half-life reported in children younger than two years of age. Atropine may cause asystole, chest pain, atrial arrhythmias, or transient atrioventricular dissociation, among others [37,76,77,78,79].

Bisoprolol, a class IIa AAD, is not available for intravenous administration. The oral dosage is reported to range from 0.1 to 0.4 mg/kg per day, with a maximum dose of 20 mg per day. Bisoprolol is mainly used in heart failure therapy. However, it may be indicated in paroxysmal supraventricular tachyarrhythmias in conjunction with heart failure. Bisoprolol’s half-life is reported to be up to 12 h when renal function is unimpaired. In patients with impaired renal or hepatic function, the half-life is reported to extend up to 36 h. The onset of action is reported within 2 h following oral administration, with peak action reported 2 to 4 h following administration. Bisoprolol may cause chest pain, fatigue, diarrhea, or an increase in hepatic transaminases (AST/ALT), among others [42,43,44,45].

Carvedilol, a class IIa AAD, is not available for intravenous administration. The oral dosage is reported to range from 0.075 to 0.8 mg/kg per dose, with a maximum reported dose of 50 mg per day (NOT per kg). Carvedilol is mainly used in heart failure therapy. However, it may be indicated in paroxysmal supraventricular tachyarrhythmias in conjunction with heart failure. Carvedilol’s half-life is reported to be up to 3.6 h. The onset of action is reported within 1 h following administration, with an earlier alpha blocking effect (within 30 min) and later onset beta blocking effect. Carvedilol may cause arterial hypotension, dizziness, fatigue, weight gain, or hyperglycemia, among others [46,47,48,49,50,51].

For digoxin, a class IId AAD, the intravenous dose is reported to range from 0.01 to 0.015 mg/kg per dose, administered every 8 h. The maximum dosage is reported to be up to 0.2 mg/dose. Intravenous administration aims for a blood target level of 1 to 2.5 nmol/L or 0.78 to 1.95 ng/mL. For oral administration, the dosage is reported based on body weight (BW), as well as based on body surface area (BSA). The BW dosage is reported to range from 0.01 to 0.017 mg/kg per dose, administered every 8 to 12 h. The maximum dosage based on BW is reported to be up to 0.017 mg/kg/dose. Based on BSA, the dosage is reported to be 0.2 mg/m^2^/dose, administered every 8 to 12 h. Common indications for digoxin include supraventricular tachyarrhythmias, including atrial fibrillation or atrial flutter for ventricular rate control. It is also used to treat prenatal tachyarrhythmias. Digoxin’s half-life is dependent on age and cardiac and renal function. The half-life is reported to be up to 170 h in preterm born infants. For the pediatric cohort, the average half-life is reported to be 18 to 36 h. The onset of action is reported within 2 h following oral administration and within 5 to 60 min following intravenous administration. Digoxin may cause arrhythmias, asystole, heart block, ST segment depression, facial edema, or premature ventricular contractions, among others [22,24,26,35,72,80,81,82,83].

For esmolol, a class II AAD, the intravenous dosage is reported to range from 0.01 to 0.5 mg/kg as an initial bolus over one minute, followed by continuous infusion of 10 to 500 mcg/kg/min. The maximum dose is reported to be 0.5 mg/kg over one minute for the initial bolus dose and up to 500 mcg/kg/min for continuous infusion. Esmolol is not available for oral administration. Furthermore, esmolol continuous infusion has been well documented up to 48 h. Data on longer continuous use currently remain scarce. Therefore, the duration of use should be limited to 48 h, transitioning the patient to an alternative betablocker. Common indications for esmolol include supraventricular tachyarrhythmias or postoperative junctional ectopic tachycardia. Esmolol’s half-life is reported to be up to 5 min. The onset of action is reported to be within 2 to 10 min following intravenous administration. The duration of action is reported to be up to 30 min. Esmolol may cause arterial hypotension, confusion, nausea, vomiting, headache, agitation, or infusion site reactions including inflammation or induration, among others [24,31,36,52,53,54,55,56,57].

For isoprenaline, a class IIb AAD, the intravenous dosage is reported to range from 0.05 to 2.0 mcg/kg/min. The maximum dosage is reported to be up to 2.0 mcg/kg/min. Isoprenaline is not available for oral administration. Common indications for isoprenaline include long QT syndrome with torsades or atrioventricular block. Isoprenaline’s half-life is reported to be up to 5 min, with immediate onset following intravenous administration. The effect duration is reported to be up to 15 min. Isoprenaline may cause flushing, angina pectoris, arterial hyper- or hypotension, palpitations, restlessness, arrhythmias, or seizures, among others [84,85,86].

Landiolol, a novel substance with rapid onset beta blocking properties (class IIa), has not yet been approved for pediatric use. Therefore, dosage recommendations are still lacking. Currently available reports on landiolol use in pediatric patients demonstrate favorable results for tachyarrhythmia treatment in pediatric patients [58]. Further studies will be needed for landiolol to be included in pediatric arrhythmia guidelines. Current reports on safety and efficacy in pediatric patients are promising. Landiolol is available for intravenous administration only. The dosage is reported to range from 1 to 5 mcg/kg/min iv. It may then be uptitrated to 40 mcg/kg/min iv, based on efficacy. Landiolol’s half-life is reported to be up to 4 min, with an onset of action reported within 1 min following iv administration. Landiolol may cause arterial hypotension, bradycardia, dizziness, or headache, among others [58,59,60,61,62].

For metoprolol, a class II AAD, the dosage is reported to be 0.1 mg/kg per intravenous dose. Up to three doses are reported to be administered every 5 min based on efficacy. Bolus administration may be followed by continuous infusion ranging from 1 to 5 mcg/kg/min. The maximum dosage is reported to be up to three consecutive doses of 0.1 mg/kg or 5 mcg/kg/min for continuous infusion. For oral administration, the dosage is reported to range from 0.5 to 6 mg/kg/day, administered every 8 to 12 h. The maximum dosage is reported to be up to 2 mg/kg/dose. Common indications for metoprolol include paroxysmal supraventricular tachycardias, especially in conjunction with heart failure. Metoprolol’s half-life is reported to be up to 10 h in neonates and up to 4 h in adults. The half-life is strongly influenced by CYP2D6 metabolism. The onset of action is reported within 20 min following iv administration or 1 h following immediate release oral administration. Metoprolol may cause arterial hypotension, bradycardia, fatigue, depression, or vertigo, among others [41,44,48,51,63,64,65].

For nadolol, a class IIa AAD, the oral dosage is reported to range from 0.5 to 5.0 mg/kg per dose, administered once daily. The maximum dosage is reported to be up to 5 mg/kg/day or 240 mg/day (NOT per kg). There is no preparation of nadolol available for intravenous administration. Common indications for nadolol include long QT syndrome or paroxysmal supraventricular tachycardias. Nadolol’s half-life is reported to be up to 5 h in infants, 16 h in children, and 7 h in adolescents. Peak serum levels are reported within 3–4 h following administration. The effect duration is reported to be up to 24 h. Nadolol may cause arterial hypotension, bradycardia, edema, palpitations, Raynaud’s phenomenon, depression, or sedation, among other things [20,22,41,44,64,65,66,67,68,69,70,71].

For propranolol, a class IIa AAD, the dosage is reported to range from 0.01 to 0.3 mg/kg per intravenous dose, administered every 3 h. The maximum dosage is reported to be up to 1.0 mg/day (NOT per kg) in children below one year of age. For children above one year of age, the maximum dosage is reported to be up to 5 mg/day (NOT per kg). For oral administration, the dosage is reported to range from 0.5 to 16 mg/kg/day, administered every 6 to 8 h. The maximum oral dosage is reported to be 320 mg/day (NOT per kg). Common indications for propranolol include supraventricular tachyarrhythmias or long QT syndrome. Propranolol’s half-life is reported to be up to 6.5 h, with the onset of action reported within 2 h following oral administration and within 5 min following iv administration. Propranolol may cause sleep disorder (most prominently in infants), agitation, bronchitis, or cold extremities, among other things [2,20,24,27,31,41,43,44,54,64,69,70,72,73,74].

Class III antiarrhythmic drugs

For amiodarone, a class IIIa AAD, the dosage is reported as 5 mg/kg for the initial iv dosage. The maximum dosage is reported to be up to 15 mg per dose or up to 30 mg/kg per day. If using continuous iv dosing, the initial dosage is reported to range from 7 to 20 mg/kg per hour, not to exceed a maximum daily dose of 30 mg/kg, as aforementioned. If used orally, the dosage is reported to range from 3 to 15 mg/kg per day, with a reported maximum dose of 8 mg/kg per dose or 200 mg per day (NOT per kg). When administering amiodarone loading maximum is 400mg per day (NOT per kg). Common indications for amiodarone in pediatric patients include life-threatening ventricular tachycardia or ventricular fibrillation according to current ERC guidelines, hemodynamically stable wide complex tachycardia, or drug refractory supraventricular tachyarrhythmias [40]. Its use is also recommended for rhythm control in atrial fibrillation (AF) [102]. However, AF is uncommon in pediatric patients. When administered orally, amiodarone’s half-life is reported to be up to 40 days. Following iv administration, the reported half-life ranges from 9 to 36 days. The onset of action is reported to range from 2 days to 3 weeks when administered orally and is reported to occur within hours when administered iv. Amiodarone may lead to arterial hypotension, nausea, vomiting, or epithelial keratopathy in up to 99% of patients when administered over long periods of time. It also exhibits pulmonary toxicity, and routine follow-up with regard to side effects is recommended in patients on long-term amiodarone. Baseline parameters should be established when starting amiodarone therapy. These should include electrolytes, urea and creatinine, liver function tests (LFT´s), thyroid function tests (TFT´s), chest X-ray, eye examination, and 12-lead ECG. For long term follow-up, it is recommended to perform LFTs, TFTs, respiratory function assessment, and visual function assessment every six months [2,25,27,35,40,96,97,98,99].

Sotalol, a class III AAD, dosage is reported ranging from 0.5 to 2.0 mg/kg intravenously per dose, administered every 12 hours. A single bolus dose is reported to be followed by continuous infusion ranging from 1 to 6 mg/kg/h. Sotalol is also reported for use as continuous infusion only (no prior bolus). Maximum dosage is reported up to 120 mg/dose (NOT per kg) or 6 mg/kg/h. For oral administration dosage reports are more complex. Dosage may be based on body weight (BW) or body surface area (BSA). Based on BW dosage is reported ranging from 2 to 26 mg/kg/day, administered every 8 to 12 hours. However, there are different target doses reported depending on patient age. In children aged 1 month to 6 years, target dose is reported up to 6 mg/kg/day. For children older than 6 years, target dosage is reported up to 4 mg/kg/day. Maximum dosage based on BW is reported up to 26 mg/kg/day or 480 mg/day (NOT per kg). Based on BSA, dosage is reported up to 90 mg/m^2^/day. However, in children younger than 2 years, dosage is recommended to be corrected by an age factor, downgrading the absolute dose. In the first week the age factor is reported as 0.3, 1 week to 1 month of age the factor is reported as 0.6. For infants older than one month, the factor is reported to increase up to 0.97 at 20 months of age. Usually BSA based dose is corrected by the above mentioned factors (e.g. 30 mg/m^2^ × 0.3 in a 5 days old newborn). For the first week of life reported dosage ranges from 60 to 200 mg/m^2^/day. Maximum dosage is reported up to 60 mg/m^2^/day or 320 mg/day (NOT per kg). Common indications for sotalol include supraventricular and ventricular tachyarrhythmias. Sotalol half life is reported up to 9 hours, with prominent prolongation in patients with renal failure. Onset of action is reported within 2 hours following oral administration, and ranging from 5 to 10 minutes following iv administration. Sotalol may cause bradycardia, chest pain, dyspnea, fatigue, headaches, or weakness, among others [22,24,26,30,63,103,104,105,106,107,108,109,110,111,112,113].

Class IV antiarrhythmic drugs

For diltiazem, a class IVa AAD, the dosage is uniformly reported to be 0.25 mg/kg per dose, with a maximum dose of 20 mg per dose. When administered intravenously, the initial dose is reported to range from 0.05 to 0.25 mg/kg per hour, not to exceed 15 mg per hour (NOT per kg). When administered orally, the initial dosage is reported to range from 1.0 to 3.0 mg/kg per dose, with a maximum dose of 3.5 mg/kg per day or 180 mg per day (NOT per kg). Common indications for diltiazem include supraventricular tachyarrhythmias. Calcium channel blockers should not be used in children below one year of age due to possible hemodynamic impairment. Diltiazem’s half-life is reported to be up to 7 h following oral administration and up to 5 h following intravenous administration. Diltiazem may cause headache, gingival hyperplasia, constipation, peripheral edema, or arterial hypotension. The latter is reported as a potential fatality in infants younger than one year of age. Therefore, its use should be restricted to children one year of age or older [92,94,95].

For verapamil, a class IVa AAD, the dosage is reported to range from 0.1 to 0.3 mg/kg intravenously (bolus dose) over ten minutes, followed by continuous infusion of 5 mcg/kg/min. The maximum dosage is reported to be up to 15 mg/dose (NOT per kg). However, there is no maximum daily dose reported. For oral administration, the dosage is reported to range from 2 to 9 mg/kg/day, administered every 8 h. The maximum oral dosage is reported to be up to 480 mg/day (NOT per kg). Verapamil should not be used in children below 1 year of age. Common indications for verapamil include supraventricular tachyarrhythmias or Belhassen VT. Verapamil’s half-life is reported up to 7 h following oral administration and up to 5 h following iv administration. Verapamil may cause headache, gingival hyperplasia, constipation, peripheral edema, or arterial hypotension. The latter is reported to be potentially fatal in infants younger than one year. Therefore, its use should be restricted to children >1 year of age [63,87,88,89,90,91,92,93].

## 4. Discussion

Indications for AAD therapy in children are very well outlined in official consensus documents endorsed by the European Heart Rhythm Association (EHRA) and the Association for European Pediatric and Congenital Cardiology (AEPC) [2,3]. However, the dosage of AADs in children, though homogeneous for some drugs, is reported to be somewhat heterogeneous. For example, for adenosine, which is exclusively used in the acute treatment of arrhythmias and exists in iv dosage form only, the pediatric dosage is uniformly reported as 0.1 mg/kg per dose. For other substances, e.g., sotalol, the reported dosage ranges quite extensively. As for iv sotalol, the reported dosage ranges from 2 to 26 mg/kg/day. Such ranges lead to difficulties in clinical practice. This seems even more true for smaller centers where experience in pediatric arrhythmias and electrophysiology may be limited. Future guidelines on the treatment of arrhythmias in children may include dosage recommendations based on task force consensus agreement. A European multicenter study evaluating the dosage of antiarrhythmic drugs in the treatment of pediatric arrhythmias may aid such recommendations.

Most centers may have their “favorites” among AADs which are routinely used. As for our center, we routinely use metoprolol, propranolol, esmolol, flecainide, sotalol and amiodarone. Our dosing approach for the above-mentioned drugs is as follows:

Metoprolol: initial dosage of 0.1 mg/kg over 1 h iv; uptitrate to 0.3 mg/kg over one hour iv, if needed. The maximum iv dose is 5 mg/kg/day. For rate control in the acute treatment of SVT, we prefer esmolol over metoprolol or propranolol. When using betablockers for long-term treatment, however, we routinely use metoprolol at an initial dosage of 0.5 to 1 mg/kg per dose p.o., once daily (extended release preparation). Below one year of age, we prefer propranolol over metoprolol.

Propranolol: initial dosage of 0.05 mg/kg over 10 min iv, administered every 6 h, if needed. Maximum iv dose: 1 mg per dose in infants, 3 mg per dose in children up to 6 years of age, or 60 mg per day (NOT per kg!). When using propranolol for long-term treatment (e.g., long QT syndrome or SVT), we initiate therapy as 0.5 to 1 mg/kg per day p.o., administered every 6–8 h. We uptitrate to a maximum of 16 mg/kg per day.

Esmolol: initial dosage of 100 mcg/kg over 1 min iv bolus, followed by 50 mcg/kg/min continuous infusion. We uptitrate continuous infusion to the effect (max. 500 mcg/kg/min). For rate control in the acute treatment of SVT, esmolol is our first choice.

Flecainide: initial dosage of 50 mg/m^2^ per day p.o. in infants younger than one year, administered every 8–12 h. In children one year of age or older, we use 100 mg/m^2^ per day p.o. every 8–12 h. Flecainide is our first choice for the long-term treatment of SVT, with a special emphasis on Wolff–Parkinson–White syndrome. When testing for Brugada syndrome, we use ajmaline.

Sotalol: initial dosage of 6 mg/kg per day p.o. in children younger than six years, administered every 8 h. We uptitrate to a maximum 15 mg/kg per day or 320 mg per day (NOT per kg!). In children six years of age and older, we initially administer 4 mg/kg per day p.o. every 8 h. We use sotalol in patients with congenital heart disease (e.g., Tetralogy of Fallot or complete AV canal) for the long-term treatment of SVT. We do not use sotalol iv preparations.

Amiodarone: initial dosage of 5 mg/kg over 5 to 30 min of iv bolus, followed by continuous infusion of 5–15 mg/kg per day. Maximum of 15 mg/kg per day or 1200 mg per day (NOT per kg!). Amiodarone iv is our first choice in treating wide complex tachycardias, especially in children with congenital heart defects (CHD). In patients with CHD, we use amiodarone for long-term treatment at an initial dose of 5 mg/kg per day p.o., administered every 8 h. Maximum oral dosage: 12 mg/kg per day or 1200 mg per day (NOT per kg!). However, long-term adverse effects of amiodarone should be considered. We prefer sotalol for long-term treatment in select patients.

We use other substances on a case-by-case basis. As for nadolol, this substance is not available in Austria. Experience with certain substances may vary among centers. However, experience is important, as it leads to confidence in the use of certain drugs. Substances that are barely ever in use are typically associated with insecurity regarding the correct initial dosing and uptitration throughout therapy. Dosage failure may result.

Furthermore, the availability of substances and their approval by authorities play a crucial role when it comes to AAD therapy in children. While some substances (e.g., propranolol) may be widely available, others such as nadolol or ajmaline are not. The availability of substances will vary between centers and countries. Most likely, the experience will be high with substances that have been available for a long time at a certain center.

Therefore, we would recommend that centers establish their own specific protocols with regard to AAD therapy in children suffering from arrhythmias. Whenever possible, substances that are recommended within official guideline documents (e.g., nadolol in long QT syndrome) should be used. In case these substances may not be available, which unfortunately still remains true, e.g., nadolol in some European countries, local consensus should be achieved as for which substance may serve as a replacement. Local center-specific protocols in this regard should prevent the arbitrary use of different substances and provide a uniform approach to AAD therapy.

When treating patients with antiarrhythmic drugs, physicians must be aware of overdosing or intoxications. In the current era of COVID, depression and hospitalizations of children, and even more so of adolescents, due to intoxications or suicidal attempts have increased [114,115]. AADs are highly effective but also potentially lethal drugs, and the recognition of either intentional or unintentional intoxication is of the utmost importance to establish sufficient therapy quickly. As with digoxin, overdose may be treated by administering digoxin-specific antibody fragments (FAB) [116]. Usually, the dosage of FAB is based on serum digoxin levels. An approximate dose can be calculated as “serum digoxin level (nmol/L) × body weight (kg) × 0.3″ [117,118]. In class I AAD overdoses (i.e., with propafenone or flecainide), sodium bicarbonate (e.g., start 1 mmol/kg iv; high doses needed!), isoproterenol (e.g., up to 2 mcg/kg/min iv), lipid emulsions (e.g., intravenous lipid emulsion 20% 1–1.5 mL/kg iv bolus over 1 min, followed by continuous infusion of 0.25–0.5 mL/kg/min), or calcium-gluconate (e.g., calcium gluconate 10% 0.5 mL/kg over 30 min iv) are recommended [119,120,121,122]. For betablocker intoxications, activated charcoal (1 g/kg, max. 50 g), calcium-gluconate (e.g., calcium gluconate 10% 0.25–0.5 mL/kg over 30 min iv), isoproterenol e.g., (up to 2 mcg/kg/min iv), atropine (e.g., 0.02–0.04 mg/kg iv), or glucagon (e.g., 0.1 mg/kg iv) are recommended, among other substances [123,124]. The consultation of poison control services is strongly recommended when treating patients for AAD overdose.

More recently developed substances such as landiolol or ivabradine may be introduced in future guidelines on pediatric arrhythmias. Currently, their use and dosing remain off-label and subject to future studies. Both landiolol and ivabradine, however, have been reported to show promising results with regard to tachyarrhythmia therapy in children.

Within this manuscript, we aimed to summarize the dosage for what we consider the most commonly used AADs. All provided dosages must be carefully checked before use in patients.

## 5. Conclusions

Within this review article, we focused on the most commonly used AADs in pediatric arrhythmias and aimed to provide an overview on the dosage, as reported in the literature. We encourage centers to establish their own protocols for AAD dosing in children to provide a center-specific uniform approach and prevent dosing failure whenever possible.

## Figures and Tables

**Table 1 children-10-00847-t001:** Dosage of Vaughn–Williams classification 0, Ia, and Ib drugs in children [2,5,6,7,8,9,10,11,12,13,14,15,16,17,18,19,20,21].

Antiarrhythmic Agent & Class	Iv	MAX	INTERVAL	po	MAX	INTERVAL
Vaughn–Williams Class 0						
**Ivabradine**						
Weight-based				0.05 mg/kg/ED <40 kg	Max. 0.2 mg/kg/ED * Max. 0.3 mg/kg/ED ** Or max. 7.5 mg/ED (NOT per kg!)	12 h
	* 6 months to 1 year of age; ** >1 year of age					
**Vaughn–Williams Class 1a**						
**Ajmaline ***						
Weight-based	0.5–1 mg/kg/ED Cont.: 0.5–1 mg/kg/h	Max. 50 mg/ED or 8 mg/kg/d or 1.200 mg/d	N.A. Max. 10 mg/min!			
* Continuous ECG monitoring during the application of Ajmaline is required! Stop application if: ≥25% prolongation of QRS or QT duration or ≥50% prolongation of PQ duration.						
**Quinidine Sulfate**						
Weight-based				15–30 mg/kg/day	Max. 60 mg/kg/d	6 h
BSA-based				900 mg/m^2^/d	N.A.	6 h
**Quinidine Gluconate**						
Weight-based	30 mg/kg/d	N.A.	5–6 h	20–60 mg/kg/day	Max. 60 mg/kg/d	8 h
BSA-based	900 mg/m^2^/d	N.A.	5–6 h			
**Vaughn–Williams Class 1b**						
**Lidocaine**						
Weight-based	1 mg/kg/ED Cont.: 20–50 mcg/kg/min	1 mg/kg/ED 50 mcg/kg/min	Loading ED followed by cont. iv			
**Mexiletine**						
Weight-based				1.4–5.0 mg/kg/ED	Max. 8 mg/kg/ED or 1200 mg/d	8–12 h

Leg.: BSA = body surface area, ED = enddose, cont. = continuous, min = minute, h = hour, d = day, a = year, interval is given as number of hours between single applications.

**Table 2 children-10-00847-t002:** Dosage of Vaughn–Williams classification Ic drugs in children [2,22,23,24,25,26,27,28,29,30,31,32,33,34,35,36,37,38,39,40].

Antiarrhythmic Agent & Class	iv	MAX	INTERVAL	po	MAX	INTERVAL
Vaughn–Williams Class Ic						
**Flecainid**						
Weight-based	0.2–2.0 mg/kg/ED Cont.: 0.2–0.5 mg/kg/h	max. 200 mg/d	12 h	1–7 mg/kg/ED	max. 200 mg/d or 8 mg/kg/d	8 h/12 h
BSA-based				<6 mo: 50 mg/m^2^/d >6 mo: 100 mg/m^2^/d		12 h 12 h
**Propafenone**						
Weight-based	0.2–2.0 mg/kg/ED in 2 h Cont.: 4–20 mcg/kg/min	2 mg/kg/ED in 2 h 20 mcg/kg/min	Loading ED followed by cont. iv	6–20 mg/kg/d	20 mg/kg/d	8 h/12 h
BSA-based				150–600 mg/m^2^/d	max. 900 mg/d	8 h/12 h

Leg.: BSA = body surface area, ED = enddose, cont. = continuous, min = minute, h = hour, d = day, a = year, interval is given as number of hours between single applications.

**Table 3 children-10-00847-t003:** Dosage of Vaughn–Williams classification IIa drugs in children [2,20,22,24,27,31,36,41,42,43,44,45,46,47,48,49,50,51,52,53,54,55,56,57,58,59,60,61,62,63,64,65,66,67,68,69,70,71,72,73,74].

Antiarrhythmic Agent & Class	iv	MAX	INTERVAL	po	MAX	INTERVAL
Vaughn–Williams Class IIa						
**Atenolol**						
Weight-based	0.05–0.2 mg/kg/ED	max. 2.5 mg/kg/ED	12 h	0.3–1.0 mg/kg/d	max. 2.0 mg/kg/d or 100 mg/d	12 h/24 h
**Bisoprolol**						
Weight-based				0.1–0.4 mg/kg/d	max. 10–20 mg/d	24 h
**Carvedilol**						
Weight-based				0.075–0.8 mg/kg/ED	max. 25–50 mg/d	8–12 h
**Esmolol**						
Weight-based	Bolus: 0.01–0.5 mg/kg in 1 min Cont.: 10–500 mcg/kg/min	0.5 mg/kg in 1 min 500 mcg/kg/min	Bolus followed by cont. infusion			
**Landiolol**						
Weight-based	Cont.: 1–5 mcg/kg/min	Max. 40 mcg/kg/min	cont. iv			
**Metoprolol**						
Weight-based	0.1 mg/kg/ED Cont.: 1–5 mcg/kg/min	3 consecutive ED 5 mcg/kg/min	Repeat ED every 5 min, followed by cont. iv	0.5–6 mg/kg/d	max. 1–2 mg/kg/ED	8 h/12 h
**Nadolol**						
Weight-based				0.5–5.0 mg/kg/ED	max. 2.5 mg/kg/d or 240 mg/d	24 h
**Propranolol**						
Weight-based	0.01–0.3 mg/kg/ED	<1a: max. 1 mg/d >1a: max. 3–5 mg/d	3 h	0.5–16 mg/kg/d	max. 240–320 mg/d	6 h/8 h

Leg.: BSA = body surface area, ED = enddose, cont. = continuous, min = minute, h = hour, d = day, a = year, interval is given as number of hours between single applications.

**Table 4 children-10-00847-t004:** Dosage of Vaughn–Williams classification IIb, IIc, IId, and IIe drugs in children [2,3,4,22,24,26,35,37,72,75,76,77,78,79,80,81,82,83,84,85,86].

Antiarrhythmic Agent & Class	iv	MAX	INTERVAL	po	MAX	INTERVAL
Vaughn–Williams Class IIb						
**Isoprenaline**						
Weight-based	0.05–2.0 mcg/kg/min	2 mcg/kg/min	Cont. iv			
**Vaughn–Williams Class IIc**						
**Atropine**						
Weight-based	0.01–0.04 mg/kg/ED min. 0.05–0.1 mg/ED	max. 0.5–1.0 mg/kg/ED	8 h/12 h			
**Vaughn–Williams Class IId**						
**Digoxin ***						
**Weight-based**	0.01–0.015 mg/kg/ED Blood level target: 1–2.5 nmol/L 0.78–1.95 ng/mL	max. 0.2 mg/ED	8 h	0.01–0.017 mg/kg/ED	0.017 mg/kg/ED	8 h/12 h
**BSA-based**				0.2 mg/m^2^		8 h/12 h
	* Changing Digoxin from oral to iv; dose must be reduced by 20–25%. Lower doses are needed in patients with renal failure. Total digitalizing dose is optional and is not presented within this table.					
**Vaughn–Williams Class IIe**						
**Adenosine**						
Weight-based	0.1 (0.2/0.3) mg/kg/ED <1a: 0.15 mg/kg/ED	max. 12 mg/ED	3 consecutive ED			

Leg.: BSA = body surface area, ED = enddose, cont. = continuous, min = minute, h = hour, d = day, a = year, interval is given as number of hours between single applications.

**Table 5 children-10-00847-t005:** Dosage of Vaughn–Williams classification III and IV drugs in children [2,25,27,35,40,63,87,88,89,90,91,92,93,94,95,96,97,98,99].

Antiarrhythmic Agent & Class	iv	MAX	INTERVAL	po	MAX	INTERVAL
Vaughn–Williams Class III						
**Amiodarone**						
Weight-based	5 mg/kg/ED 7–20 mcg/kg/min	max. 15 mg/kg/ED max. 10–30 mg/kg/d	8 h/12 h/24 h	3–15 mg/kg/d	max. 8 mg/kg/ED or 200 mg/d	8 h/12 h/24 h
**Sotalol**						
Weight-based	0.5–2.0 mg/kg/ED Cont.: 1–6 mg/kg/h	Max. 120 mg/ED Cont. max. 6 mg/kg/h	12 h	2–26 mg/kg/d Target dose 1 mo–6 y: 6 mg/kg/d >6 y: 4 mg/kg/d	max. 26 mg/kg/d or 200–480 mg/d	8 h/12 h
BSA-based				90 mg/m^2^/d 1st WOL: 60–200 mg/m2/d <2 years of age corrected for age **	60 mg/m^2^/d or 320 mg/d	8 h
**Vaughn–Williams Class IV**						
**Diltiazem**						
Weight-based	0.25 mg/kg/ED 0.05–0.25 mg/kg/h	max. 20 mg/ED 5–15 mg/h	1 ED followed by cont. infusion	1.0–3.0 mg/kg/ED	max. 3.5 mg/kg/d or 180 mg/d	8 h
**Verapamil**						
Weight-based	0.1–0.3 mg/kg/ED Cont.: 5 mcg/kg/min	max. 5–15 mg/ED	Loading ED over 10 min, followed by cont. iv	2–9 mg/kg/d	max. 120–480 mg/d	8 h

Leg.: BSA = body surface area, ED = enddose, cont. = continuous, min = minute, h = hour, d = day, a = year, interval is given as number of hours between single applications. ** <2 years of age; the manufacturer recommends dosage reduction based on an age factor determined. In infants with BSA <0.33 m^2^, a larger drug exposure (larger AUC) and greater pharmacologic effects for a given dose per m^2^ were observed compared to infants with BSA ≥0.33 m^2^. WOL = week of life.

**Table 6 children-10-00847-t006:** Common antiarrhythmic drugs in children and their indications, preparations, and mechanism of action.

Drug	V-W Class	Main Indications	Preparations	Mechanism of Action	Pharmacokinetics
Ivabradine	0	IST, AT, JET	po	HCN channel blocking	HL 6 h; peak level 1 h (2 h with food)
Ajmaline	Ia	BrS testing	iv	Na^+^ channel blocking (imt)	HL up to 96 min
Quinidine	Ia	BrS, SQTS	po, iv	Na^+^ channel blocking (imt)	HL up to 8 h; onset of action within 3 h
Lidocaine	Ib	VT	iv	Na^+^ channel blocking (fast)	HL 2 h; onset of action 90 s
Mexiletine	Ib	LQTS-3	po	Na^+^ channel blocking (fast)	HL 12 h; onset of action within 120 min
Flecainide	Ic	SVT, WPW	po, iv	Na^+^ channel blocking (slow)	HL 8 h (up to 29 h in newborns)
Propafenone	Ic	SVT, VT, AT, MAT	po, iv	Na^+^ channel blocking (slow)	HL up to 32 h
Atenolol	IIa	LQTS, SVT	po	Beta blocking (selective)	HL up to 35 h; onset of action 1 h
Bisoprolol	IIa	SVT	po	Beta blocking (selective)	HL 12 h; onset of action 2 h; peak level 2–4 h
Carvedilol	IIa	SVT	po	Beta blocking (non-selective)	HL 3.6 h; onset of action 1 h
Esmolol	IIa	SVT	iv	Beta blocking (selective)	HL 5 min; onset 2 min & duration of action 30 min
Landiolol	IIa	SVT	iv	Beta blocking (selective)	HL 4 min; onset of action within 1 min
Metoprolol	IIa	SVT	po, iv	Beta blocking (selective)	HL 10 h; onset of action 20 min (iv)/1 h (po)
Nadolol	IIa	LQTS, SVT	po	Beta blocking (non-selective)	HL up to 16 h; peak level 4 h; duration 24 h
Propranolol	IIa	LQTS, SVT	po, iv	Beta blocking (non-selective)	HL 6.5 h; onset of action 5 min (iv)/2 h (po)
Isoprenaline	IIb	TdP in LQTS, BrS	iv	Beta blocking (non-selective)	HL 5 min; immediate onset; duration up to 15 min
Atropine	IIc	Brady, BB/CaB Intox	iv	M_2_ receptor inhibitor	HL 10 h (longest in children <2 years)
Digoxin	IId	SVT, AF/AFL, fetal	po, iv	M_2_ receptor activator	HL 36 h; onset of action 5–60 min (iv)/2 h (po)
Adenosine	IIe	SVT	iv	A_1_ receptor activator	HL <10 s; immediate onset of action
Amiodarone	III	VT/VF	po, iv	K^+^ channel blocking (non-selective)	HL up to 36d; onset of action few h (iv)/2d (po)
Sotalol	III	SVT, VT	po, iv	K^+^ channel blocking (I_Kr_), Beta blocking	HL 9 h; onset of action 10 min (iv)/2 h (po)
Diltiazem	IV	SVT	po	L-type Ca^++^ channel	HL 7 h (shorter when given iv)
Verapamil	IV	SVT/ILVT	po	L-type Ca^++^ channel	HL 7 h (shorter when given iv)

Leg.: AF/AFL = atrial fibrillation/atrial flutter; AT = atrial tachycardia; MAT = multifocal atrial tachycardia; Brady = Bradycardia; BB = betablockers; BrS = Brugada syndrome; CaB = calcium channel blockers; HL = half life; h = hours; iv = intravenous; ILVT = idiopathic fascicular left ventricular tachycardia; imt = intermediate; LQTS = Long QT syndrome; min = minutes; po = per os; SQTS = Short QT syndrome; SVT = supraventricular tachycardia; TdP = Torsade-de-pointes; V-W class = Vaughn–Williams classification; VF = ventricular fibrillation; VT = ventricular tachycardia.

## Data Availability

Not applicable.

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
