# Peer review of "Antiarrhythmic Drug Dosing in Children—Review of the Literature"

_children, 2023, doi:10.3390/children10050847_

Round 1
Reviewer 1 Report
This review provides a helpful summary on the dose of antiarrhythmic agents in pediatric population. There are several suggestions for the authors to improve the manuscript.
1. Line 15-16 “With experience comes confidence in use of certain drugs and dosing”, What is the subject of this sentence?
2. The type of this article is a review, and it seems inappropriate to divide it into sections as “introduction materials and methods, results, and conclusions”.
3. Please provide references in Table 1
4. Can Table 1 be further refined? Which drugs are used primarily to treat which types of arrhythmias, and at what dose?Classification of drugs? Mechanisms? This can help readers easily find the information they need.
This review provides a helpful summary on the dose of antiarrhythmic agents in pediatric population. There are several suggestions for the authors to improve the manuscript.
1. Line 15-16 “With experience comes confidence in use of certain drugs and dosing”, What is the subject of this sentence?
2. The type of this article is a review, and it seems inappropriate to divide it into sections as “introduction materials and methods, results, and conclusions”.
3. Please provide references in Table 1
4. Can Table 1 be further refined? Which drugs are used primarily to treat which types of arrhythmias, and at what dose?Classification of drugs? Mechanisms? This can help readers easily find the information they need.
Author Response
Dear Reviewer,
thank you for your critical review of our manuscript. We appreciate your suggestions and comments and have made appropriate changes to the manuscript.
- Line 15-16 “With experience comes confidence in use of certain drugs and dosing”, What is the subject of this sentence?
We have rewritten the sentence to „To prevent potential mistakes and even out uncertainties with regard to dosing we aimed…”
- The type of this article is a review, and it seems inappropriate to divide it into sections as “introduction materials and methods, results, and conclusions”.
To explain the structure of our manuscript, Children author guidelines state “Structured reviews and meta-analyses should use the same structure as research articles“. We have changed the heading “Results” to “ Review on antiarrhythmic drug dosing in children”
- Please provide references in Table 1
References have been added. Also, according to other reviewers table 1 has been split into tables 1 to 5.
- Can Table 1 be further refined? Which drugs are used primarily to treat which types of arrhythmias, and at what dose? Classification of drugs? Mechanisms? This can help readers easily find the information they need.
Thank you for this comment on table 1. The aim of our review was to provide a clear overview on dosage of AADs in children. We felt that further refinement of the table, adding mechanism of action, pharmacokinetics or indications would, in our opinion, confuse the current table. Information regarding indications of certain drugs are provided in guideline papers and expert opinion manuscript on pediatric arrhythmias. Pharmacokinetics, classification of drugs and so forth are provided by relevant publications. The current manuscript can not take all of these topics fully into account as this would exceed the limitations of this review. In order to account for your comment we have now included a new table specifying these data. Due to other reviewers comments we have further separated table 1 into several tables and structured them according to Vaughn-Williams classification.

Reviewer 2 Report
This paper represents a nice summary of the literature regarding dosing of anti-arrhythmic medications in children. I only have a few minor suggestions that might improve readability.
1. The paper only includes a single table that ends up broken across several pages. This might be greatly improved by breading it into several (2-4) tables containing drugs of similar classes. A table should fit on a single page!
2. The abstract sells the paper short. The authors should change:
“With regard to dosage recommendations for specific drugs broad dosage ranges can be found for some substances, while recommendations for other substances are quite uniform.” to
There are rather uniform recommendations for some medications (including adenosine, Y, and Z), while there are only very broad dosage recommendations for others (like sotolol, B, and C).
“We aimed to summarize dosage ranges for antiarrhythmic drugs in children available in the literature.” To
We summarized the published dosage recommendations for antiarrhythmic drugs in children.
The sentence “With experience comes confidence in use 15 of certain drugs and dosing.” Should be deleted.
Maybe the last sentence in the abstract should be changed to: Because of wide variations in availability, regulatory approval, and experience, we encourage centers to develop their own specific protocols for pediatric antiarrhythmic drug therapy
3. The paper should not be organized listing the drugs alphabetically. Rather, drugs of similar classes could be grouped. (or drugs used for similar indications.) Alternatively, the authors could group drugs with strong data for the dosages and ones with weak data (resulting in a broad range).
4. I would encourage the authors to be stronger in emphasizing their own recommendations for appropriate doses. They can also be stronger in critiquing the data regarding different substances.
minor editing needed
This paper represents a nice summary of the literature regarding dosing of anti-arrhythmic medications in children. I only have a few minor suggestions that might improve readability.
1. The paper only includes a single table that ends up broken across several pages. This might be greatly improved by breading it into several (2-4) tables containing drugs of similar classes. A table should fit on a single page!
2. The abstract sells the paper short. The authors should change:
“With regard to dosage recommendations for specific drugs broad dosage ranges can be found for some substances, while recommendations for other substances are quite uniform.” to
There are rather uniform recommendations for some medications (including adenosine, Y, and Z), while there are only very broad dosage recommendations for others (like sotolol, B, and C).
“We aimed to summarize dosage ranges for antiarrhythmic drugs in children available in the literature.” To
We summarized the published dosage recommendations for antiarrhythmic drugs in children.
The sentence “With experience comes confidence in use 15 of certain drugs and dosing.” Should be deleted.
Maybe the last sentence in the abstract should be changed to: Because of wide variations in availability, regulatory approval, and experience, we encourage centers to develop their own specific protocols for pediatric antiarrhythmic drug therapy
3. The paper should not be organized listing the drugs alphabetically. Rather, drugs of similar classes could be grouped. (or drugs used for similar indications.) Alternatively, the authors could group drugs with strong data for the dosages and ones with weak data (resulting in a broad range).
4. I would encourage the authors to be stronger in emphasizing their own recommendations for appropriate doses. They can also be stronger in critiquing the data regarding different substances.
Author Response
Dear Reviewer,
thank you for your critical review of our manuscript. We appreciate your suggestions and comments and have made appropriate changes to the manuscript.
- The paper only includes a single table that ends up broken across several pages. This might be greatly improved by breading it into several (2-4) tables containing drugs of similar classes. A table should fit on a single page!
Thank you for this comment. We have split table 1 into tables 1 to 5, each covering certain drug classes and fitting one page.
- The abstract sells the paper short. The authors should change:
“With regard to dosage recommendations for specific drugs broad dosage ranges can be found for some substances, while recommendations for other substances are quite uniform.” to
There are rather uniform recommendations for some medications (including adenosine, Y, and Z), while there are only very broad dosage recommendations for others (like sotolol, B, and C).
“We aimed to summarize dosage ranges for antiarrhythmic drugs in children available in the literature.” To
We summarized the published dosage recommendations for antiarrhythmic drugs in children.
The sentence “With experience comes confidence in use 15 of certain drugs and dosing.” Should be deleted.
Maybe the last sentence in the abstract should be changed to: Because of wide variations in availability, regulatory approval, and experience, we encourage centers to develop their own specific protocols for pediatric antiarrhythmic drug therapy.
Thank you for these suggestions. We have changed/included the above-mentioned sentences in the abstract.
- The paper should not be organized listing the drugs alphabetically. Rather, drugs of similar classes could be grouped. (or drugs used for similar indications.) Alternatively, the authors could group drugs with strong data for the dosages and ones with weak data (resulting in a broad range).
We have rearranged the listing of drugs according to drug classes, as suggested.
- I would encourage the authors to be stronger in emphasizing their own recommendations for appropriate doses. They can also be stronger in critiquing the data regarding different substances
We have included our approach to antiarrhythmic drug dosing, citing the dosage we use and briefly naming common indications for use. We have also worked on critiquing data regarding different substances.

Round 2
Reviewer 1 Report
Thank you for the responses. I have no further comments.